# Experimental Modelling of a Solar Dryer for Wood Fuel in Epinal (France)

**Merlin Simo-Tagne** [1,2,*] , **Macmanus Chinenye Ndukwu** [3] **and Martin Ndi Azese** [4,5]

1 Nancy-Metz Academy, 54035, 2 rue Philippe de Gueldres, 54000 Nancy, France
2 LERMaB, ENSTIB, 27 rue Philippe Séguin, PO Box 1041, F-88051 Epinal, France
3 Department of Agricultural and Bioresources Engineering, Michael Okpara University of Agriculture, Umudike, Umuahia P.M.B. 7267, Abia State, Nigeria; ndukwumcu@mouau.edu.ng
4 Department of Mechanical Engineering, Otterbein University, Westerville, OH 43081, USA; azese1@otterbein.edu
5 Department of Mechanical and Aerospace Engineering, Aerospace Research Center, Ohio State University, Columbus, OH 43235, USA
* Correspondence: simotagne2002@yahoo.fr; Tel.: +33-(0)-644911921

**Abstract:** A solar dryer for wood was constructed and modelled based on the climatic condition of Epinal, France, during the summer, spring and winter seasons. The solar dryer was able to raise the temperature of the drying air by 38 °C in the spring and summer season with a global effective efficiency of 39%. Modelling of the drying of the log of wood was based on the global mass transfer coefficient and the geometric form of the log which was mostly cylindrical was considered. Validation was undertaken with the log covered with the bark. The coefficient of variations of numerical points with the experimental values given by the model was less than 5% with a mean average error and a mean relative error of 2.33% and 4.53%, respectively.

**Keywords:** Wood fuel; Fagus sylvatica; solar dryer; solar collector; modelling; experiment; Epinal

## 1. Introduction

In 2015, the domestic wood sector in France provided 15,560 direct jobs, which consisted of 19% of jobs in renewable energies [1]. Wood has a wide range of application in furniture making, housing, the making of artefacts and in energy generation for the production of biofuel and tar. Wood is also used to heat the households in temperate regions during winter seasons and in tropical regions during rainy seasons. In terms of households, about 7.8 million houses utilize wood for heating in France [2]. Wood energy currently dominates renewable energy produced in France, with a value of 40–47% of total renewable sources [1,3]. In France alone, the average yearly consumption of energy by a building is 186 $kWh_{ep}/m^2$ and 67% of this energy is utilized for heating purposes [4]. Dried wood consists of the bulk of the energy sources for the heating of these buildings and it has been reported that this consists of about 73% of the wood for domestic heating alone [3]. This is huge considering that in 2018 alone, 418,600 new buildings were constructed in France [5]. Therefore with the increase in the number of buildings constructed yearly, there is a need to seek a rapid solution to optimize the energy used for heating purposes. According to the development and energy management agency of France (ADEME), 90% of the wood products consumed for energy purposes comes as logs followed by pellets (9%), and other fuels (1%) that are reconstituted as briquettes and chips [1]. These logs of wood have to be dried to the desired moisture content for the intended purpose. Wood drying is, therefore, the process of removing a certain amount of available water from the product to improve the biological and thermos-physical properties of the dried product. Drying the wood to equilibrium state between

the product and its environment will conserve the wood materials, improve the product shelf life and the combustion properties.

During direct combustion of wood products, atmospheric pollutants (particles) are produced which depends on the age, type of the tool and moisture content of the wood. Also, the moisture state of wood affects the temperature dynamics and product yield during the pyrolysis or gasification process for the production of gas, biodiesel, tar and coal [3]. When the wood is not dry, its combustion is incomplete, slow and generates more gaseous pollutants. Wet wood has a low calorific value and, therefore, generates less heat than the same amount of wood in the dry state. Although, the low heating value (LHV) of wood varies with the species at the same moisture content higher variation is obtained when the different moisture level is considered for the same species of wood. For example, the LHV of anhydrous wood is about 5.2 kWh/kg, but this value decreases to about 2.6 kWh/kg when the moisture content is increased to 40% [3]. Therefore, it is clear that properly dried wood can produce a good amount of energy following its combustion while reducing the environmental impact generated by the process.

Some conventional wood dryers use fossil energy that delivers pollutants to the environment. This will add to environmental degradation, increase the cost of fossil fuel and the cost of the final product. Vented kilns and heat pump kilns generate 345 and 25 kg $CO_2$-eq per cubic meter of sawn timber respectively [6]. Currently, renewable energies are a good solution in all energetic strategies to protect our environment and for sustainable development. Solar energy is used for many decades to dry humid products throughout the world because it provides cheap and pollution-free energy [7]. Application of solar in drying some products gives acceptable drying kinetics [8]. However solar drying of woods takes longer time than other agro-products. It takes one to four months to dry the wood to equilibrium state using a solar dryer, but one to three years is recommended when open-air solar drying is used [9].

The energy required in drying wood using conventional solar dryers depends on the type of, solar dryer design, type of wood and thickness [10,11]. For example, 494 kWh/m$^3$ of heat energy was utilized in drying 27 mm thick pinewood (*Pinus pinaster*) from an initial moisture content of 120% to 14% wet basis using conventional kiln while 60 kWh/m$^3$ was utilized for the same purpose for drying and electrical ventilation using solar kiln driers [12]. Energy Efficiency and Conservation Authority [13] also reported energy utilization of about 3.2 GJ/m$^3$ (889 kWh/m$^3$) for a conventional vented kiln for drying wood. Simo-Tagne and Bennamoun [14] studied the amount of energy needed to dry 1 cubic meter of tropical sawn timber using a solar dryer in Central Africa regions. They obtained the values of 4.221 GJ/m$^3$ (1173 kWh/m$^3$), 3.024 GJ/m$^3$ (840 kWh/m$^3$) and 2.401 GJ/m$^3$ (667 kWh/m$^3$) respectively on Sapele, Iroko and Obeche showing that when the density of wood increases, it takes more energy to dry it [14].

Furthermore, apart from energy consumed for drying purposes, the amount of water to be extracted for drying of wood can also be influenced by the type of dryer and the volume of wood to be dried too. Bekkioui et al. [15,16] and Bentayeb et al. [11] presented a glazed solar dryer for temperate woods used in the Moroccan climate with a satisfactory performance for a small quantity of wood. However, the results obtained with the same dryer when deployed for long drying period with a large volume of wood in the industrial application was not satisfactory. Khouya and Draoui [17] presented a solar dryer for temperate woods using a latent heat energy storage in the Moroccan climate. They obtained a reduced drying time of about 47%. Khouya shows that this solar dryer can reduce the energy consumption ratio by 91%, a reduction of drying time by 62.4% with a good heat generation [18]. Lamrani et al. [19] presented an indirect hybrid solar dryer for temperate wood in the Moroccan climate and their results showed that solar collector can help to reduce annually the $CO_2$ emissions by 34%. Using a packed-bed thermal energy storage system, the payback period of the dryer system can be equal to 1.85 years [20]. Storage energy with phase change material in the hybrid solar dryer for temperate woods used under weather conditions of a site in Tangier, Morocco, permits to have a drying time not exceed 5 days during all the year [21]. Parabolic trough collector importance in the drying of

wood was showed by Lamrani et al. [22]. They achieved a maximum thermal efficiency of 76% using the solar dryer with a parabolic collector in the summer period. Luna et al. [23] presented a solar dryer with two solar collectors for drying woods under weather conditions of Mexico. The woods were stacked to form two layers (dimensions 2.16 × 1.60 × 2.5 m) separated from each other at a distance of 0.027 m. However, considering parameters such as the positioning of the wood, seasonal variations etc., on drying periods, they concluded that it will take about 250–300 h to reduce the moisture content of the wood from 0.94 kg/kg (db) to 0.12kg/kg (db). Previous numerical works on wood solar drying in central Africa showed that it is possible to construct cheap dryers of lower-income level [14,24–26]. Ndukwu et al. [27] have also carried out a detailed review of several solar dryer designs for wood in Africa.

The literatures reviewed above have given some insight into the parameters for the drying of wood using different designs of solar dryers. However, they also showed that results obtained are a function of the material of construction of the solar dryers, the type of wood dried, weather condition of the environment where the evaluation was undertaken and the seasonal variations at that period of the year. In Epinal France, there is a scarcity of literature on the drying parameters of beech wood with bark considering the two major seasons of spring and winter. The availability of such data considering the peculiar weather conditions of this area of France will help to optimize the process parameter for drying such wood for the sizeable number of wood industries located in this region of the country. Therefore, this work takes into consideration the climatic condition of Epinal, France, to dry logs (with bark) of beech wood (*Fagus sylvatica*) using wood panel solar dryer during two seasonal periods of spring and winter. Therefore, the major aim of this research is to present process parameter for the drying of a log of wood with the bark during the spring, summer and winter seasons in Epinal using a wood panel solar dryer. Modelling and validations were carried out to aid in the development of a future optimization tool for efficient dryer design for the location. Most models available for solar drying of logs of woods [28] do not consider the geometric form of the log which is mostly cylindrical. Again, the log presented in this study is modelled with the bark which is different from what is obtainable in the literature.

## 2. Materials and Methods

### 2.1. Material

The solar dryer presented in Figure 1a–f was developed, constructed and evaluated in the LERMaB Laboratory (Epinal, France). The solar collector covered a land area of 2.11 m². The high side and the low side have 1.1 m and 0.73 m respectively. The absorber plate is made with an aluminium panel and painted black. The incident solar radiation falls on the absorber plate through the transparent cover placed 5 cm above the absorber plate. The air gap was created under the absorber plate from which drying air heated by the collector circulates from the east side to the west side of the solar dryer with the help of a fan (DC-QG030148/12; maximum airflow velocity of 75 m³/h) at an average airflow of 28 m³/h. In-betweens this airflow and drying chamber, were two layers of wood panels. The west side of the solar dryer is shown in Figure 1b. Two modular openings allow you to adjust the air renewal flow in the drying chamber. The east side of our solar dryer is shown in Figure 1c with the electrical cabinet attached to control the fans. The north side of our solar dryer is presented in Figure 1d. This side can be dismantled to load our kiln with wood fuel to dry while Figure 1e shows the drying chamber. The air inlet and the air outlet of the solar collector, the air inlet and the air outlet of the drying chamber and the fans for circulating the air in the drying chamber can be distinguished. The arrangement of the three fans used to facilitate heat exchange in the drying chamber is shown in Figure 1f.

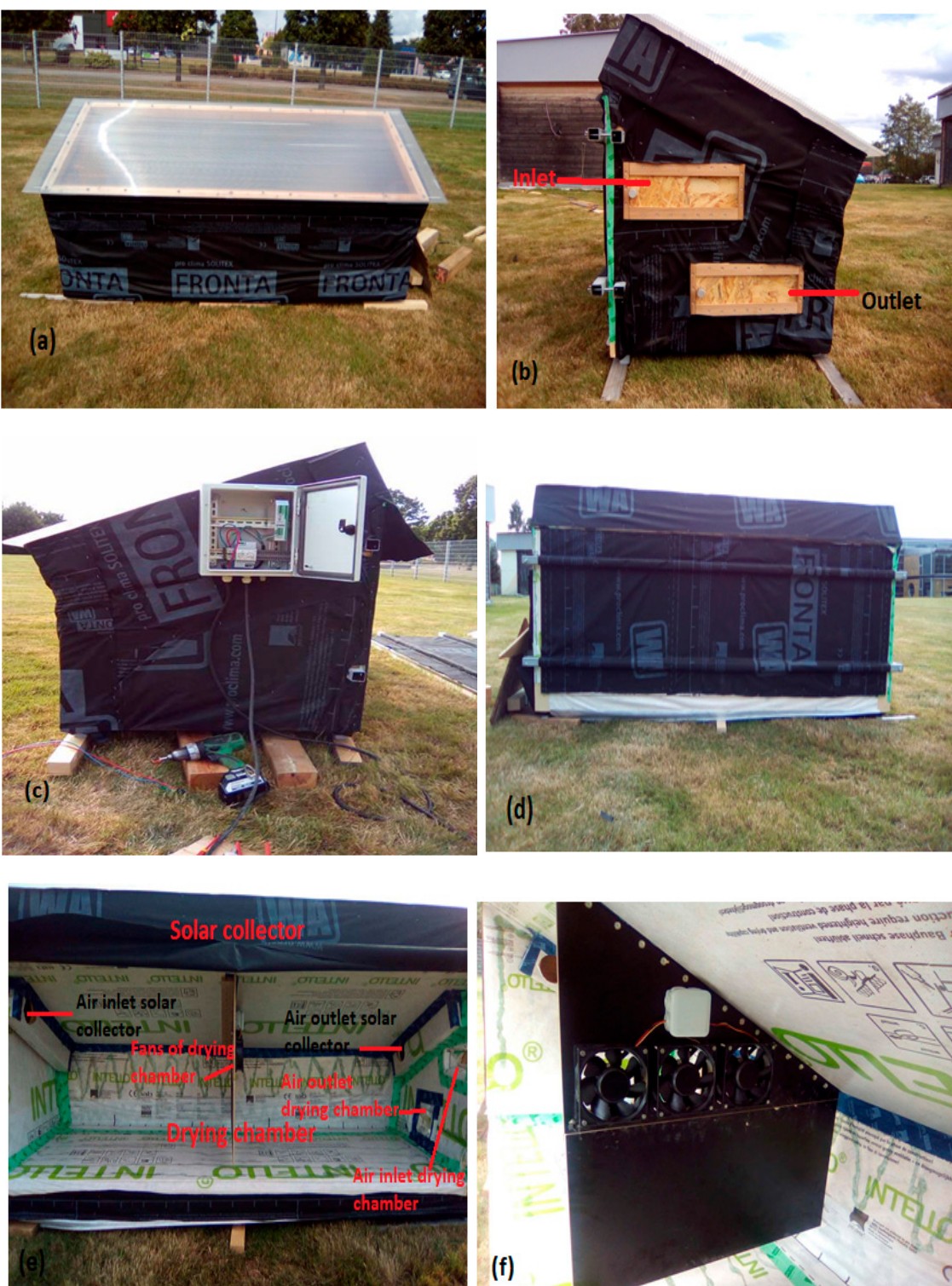

**Figure 1.** Presentation of the studied solar dryer, (**a**) south side and solar collector, (**b**) west side modular openings, (**c**) east side and electrical cabinet, (d) north side and door, (**e**) drying chamber, (**f**) fans in the drying chamber.

To protect the dryer against humidity, the interior was covered with a layer of white vapour barrier (Figure 1e–f), and the exterior with a layer of black rain barrier (Figure 1a–d). To limit the heat exchange between the interior and the exterior through the walls and the floor of our dryer, these parts

are made up of two layers of wooden panels 5 cm thick sandwiched in between, with a layer of wood fibre 5 cm thick. To view the evolution of drying, four temperature sensors, two relative humidity sensors and a mass sensor were installed. Figure 2 shows the drying arrangements with the position of the sensors and 15 logs of wood to dry. The location of the sensors made it possible to remotely monitor the drying process and automatically record the values of temperatures, relative humidity and the progress of the masses of the logs during the drying process. Pt100 sensors have been used and calibrations permit us to obtain results with a square correlation equal to 0.9999. The doors of the dryer represented by the entire north side were closed tightly during the experiment.

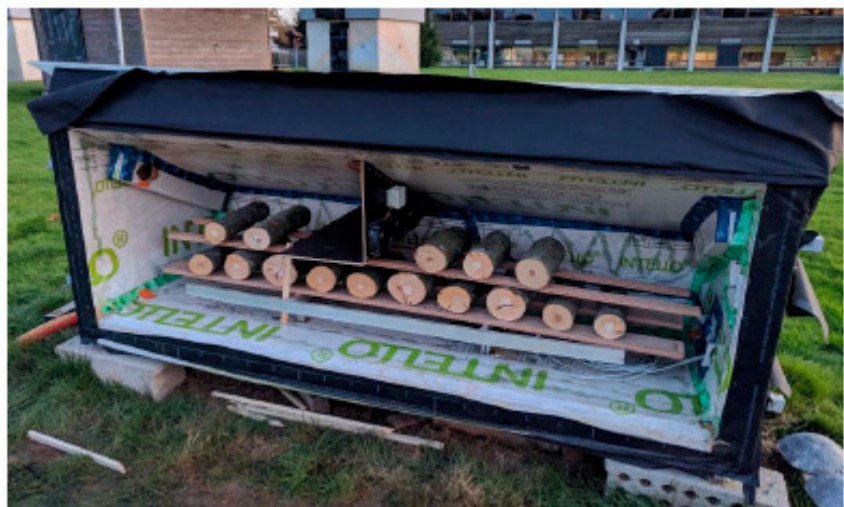

**Figure 2.** Drying chamber equipped with sensors and logs of wood to dry.

The price to build a solar dryer to this type in Epinal with 7 stere of capacity is near 668€. Using this in Epinal, the payback is after 3 years and 3 months [29]. Used in Africa, where the meteorological environment is favourable, the payback will be lower than 3 years, for a total duration of 10 years.

*2.2. Modelling*

Water in hygroscopic material is presented in two forms: bound water and free water. To model the transport of humidity in wood, it is recommended to take into account these two forms completed in vapour form [30,31]. Thus, a mass diffusion coefficient of each type of water is often modelled. In this study, the geometrical form of logs is cylindrical, completely different from those used in the literature [28]. The logs used were covered with bark, compared to other studies found in the literature. To have an idea of evolution of the moisture content of logs, we assumed that the transfer of water within the log stack can be reduced to a purely diffusive transfer. According to some literatures [32,33], Equation (1) presents the mass transfer on the wood stack:

$$-m_0 \frac{dX}{dt} = KA_b\big(X - X_{eq}\big) \tag{1}$$

To give a good idea of the physics of the process it is assuming theoretically that the process of mass transfer is not coupled with heat transfer [33]. Therefore the global mass transfer coefficient K (kg.m$^{-2}$.s$^{-1}$) is assumed to depend on the temperature of the drying air, the relative humidity of the drying air, wood stack fibre saturation point, wood stack sorption isotherm, and velocity of drying air as expressed in Equation (2) coming from [32,33]:

$$\frac{1}{K} = a_o \exp\Big(\frac{c_o}{T_a}\Big)e + b_o \exp\Big(\frac{c_o}{T_a}\Big)v_{air}^{-p} \exp\Big(-\frac{1 - RH}{X_{fsp} - X_{eq}}\Big) \tag{2}$$

where: $a_0 = 0.2265$ m.s/kg; $b_0 = 268.9$ m$^2$/kg; $c_0 = 2543.6$K; $p = 2.7158$. $v_{air}$ is the drying air velocity (m/s), e is the board thickness used (m), RH is the relative humidity of the drying air (%/100), $T_a$ is the drying air temperature in °C, $A_b$ is the total surface of the logs (m$^2$), $m_o$ is the anhydrous mass of all the samples (kg), t is the drying time (s), X is the moisture content in kg/kg, $X_{eq}$ and $X_{fsp}$ are respectively the desorption isotherm and the moisture content at the fibre saturation point which depend on the wood type [28,34,35]. The sorption isotherms of temperate wood were estimated with Equation (3a)–(3e) given by [36] as follows.

$$X_{eq} = \frac{18}{A}\left[\frac{B.RH}{1 - B.RH} + \frac{C.B.RH + 2.C.D.B^2.RH^2}{1 + C.B.RH + C.D.B^2.RH^2}\right] \tag{3a}$$

$$A = 349 + 1.29T_a + 0.0135T_a{}^2 \tag{3b}$$

$$B = 0.805 + 0.000736T_a - 0.00000273T_a{}^2 \tag{3c}$$

$$C = 6.27 - 0.00938T_a - 0.000303T_a{}^2 \tag{3d}$$

$$D = 1.93 + 0.0407T_a - 0.000293T_a{}^2 \tag{3e}$$

T in °C and RH in %/100.
$X_{fsp} = X_{eq}(RH=1)$, thus we have:

$$X_{fsp} = \frac{18}{A}\left[\frac{B}{1 - B} + \frac{C.B + 2.C.D.B^2}{1 + C.B + C.D.B^2}\right] \tag{4}$$

Using the finite differences method, Equation (1) gives:

$$X^{j+1} = \left(1 - \frac{K.A_b.\Delta t}{m_o}\right)X^j + \frac{K.A_b.\Delta t.X_{eq}^j}{m_o} \tag{5}$$

$X^{j+1}$ and $X^j$ are the moisture content at the time $t + 1$ and $t$ respectively. $\Delta t$ is the time step. The thermal efficiency of the solar dryer each day is obtained using Equation (6).

$$\eta_{th} = 100\frac{\dot{m}.C_p.(T_a - T_e)}{I_g.A_c} \tag{6}$$

$\dot{m}$ is the air mass rate in the drying chamber (kg/s), $C_p$ is the air thermal mass heat (J/(kg °C)), $T_e$ is the ambient temperature (°C), $T_a$ is the drying air temperature (°C), $I_g$ is the solar irradiation (W/m$^2$), $A_c$ is the surface of the solar collector (m$^2$).

Global effective efficiency of all solar dryer is given in Equation (7) as follows.

$$\eta_{gl} = 100\frac{m_o.C_{pb}.\sum_j\left(T_{bj} - T_{ej}\right) + (m_i - m_f).L}{\sum_j\left(t_j.I_{gj}.A_c + P_j.t_{dj}\right)} \tag{7}$$

$P_j$ is the electrical power of all fans (W), $m_o$ is the anhydrous mass of wood fuel (kg), $T_b$ is the wood fuel temperature (°C), $C_{pb}$ is the wood fuel thermal mass heat (J/(kg.°C)), $t_{dj}$ is the total duration that fans work (s), j is the hour index, $m_i$ and $m_f$ are respectively the initial and final mass of logs stack (kg), L is the latent heat of vaporization of water (J/kg).

To obtain anhydrous mass $m_o$, we used Equation (8) where $m_i$ is the initial mass of the logs at the moisture content $H_i$.

$$m_0 = \frac{m_i}{1 + H_i} \tag{8}$$

The mean absolute error (MAE) and mean relative error (MRE) were used to validate the drying kinetic model taken from the literature [19,25]. These errors are defined in Equations (9) and (10).

$$\text{MAE}(\%) = \frac{1}{N} \sum_{i=1}^{N} \left| X_{fs,exp,i} - X_{fs,theor,i} \right| \tag{9}$$

$$\text{MRE}(\%) = \frac{100}{N} \sum_{i=1}^{N} \frac{\left| X_{fs,exp,i} - X_{fs,theor,i} \right|}{X_{fs,exp,i}} \tag{10}$$

## 3. Results

Figure 3 presents the experimental differences of temperature and relative humidity between the air inlet and air outlet of solar collector obtained from 7 June 2017 and 13 June 2017 in Epinal (Spring season). From the results, the air temperature difference increases in our solar collector and peaked between 20 °C and 30 °C during the day, but during the night exterior air and interior air have the same values. The solar collector decreases the values of air relative humidity to 50% during the day, but during the night our solar collector did not have any influence on the values of air relative humidity. This is one of the challenges of solar dryers that leads to re-wetting of the product which in most cases has been overcome by the addition of thermal storage or supplementary heater to continue the drying process during the night or off-sunshine periods [37–39]. During the day, the thermal efficiency of the solar collector was estimated at 30%.

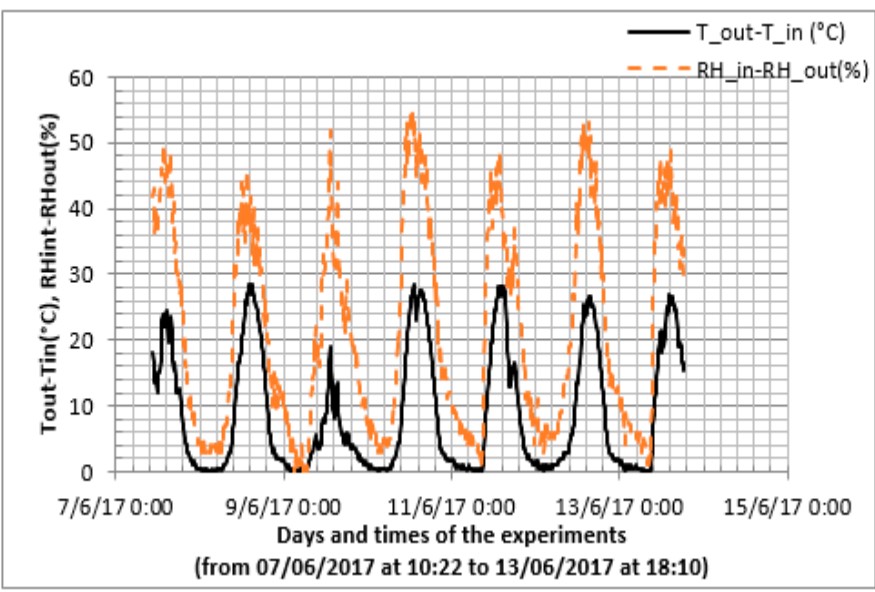

**Figure 3.** Influences of the solar collector on the characteristics of air during a week of the spring season in Epinal (door open during the experiment).

The influence of the relative humidity on ambient air temperature and drying chamber air temperature when the door is closed is presented in Figure 4a,b respectively for six weeks of the spring season in Epinal. The results were presented from 1 June 2019 to 16 July 2019. The maximum temperature of 75 °C was recorded in the kiln on the 29th June 2019 which was 38 °C higher than the corresponding ambient temperature. This shows the effectiveness of the design in raising the drying chamber temperature.

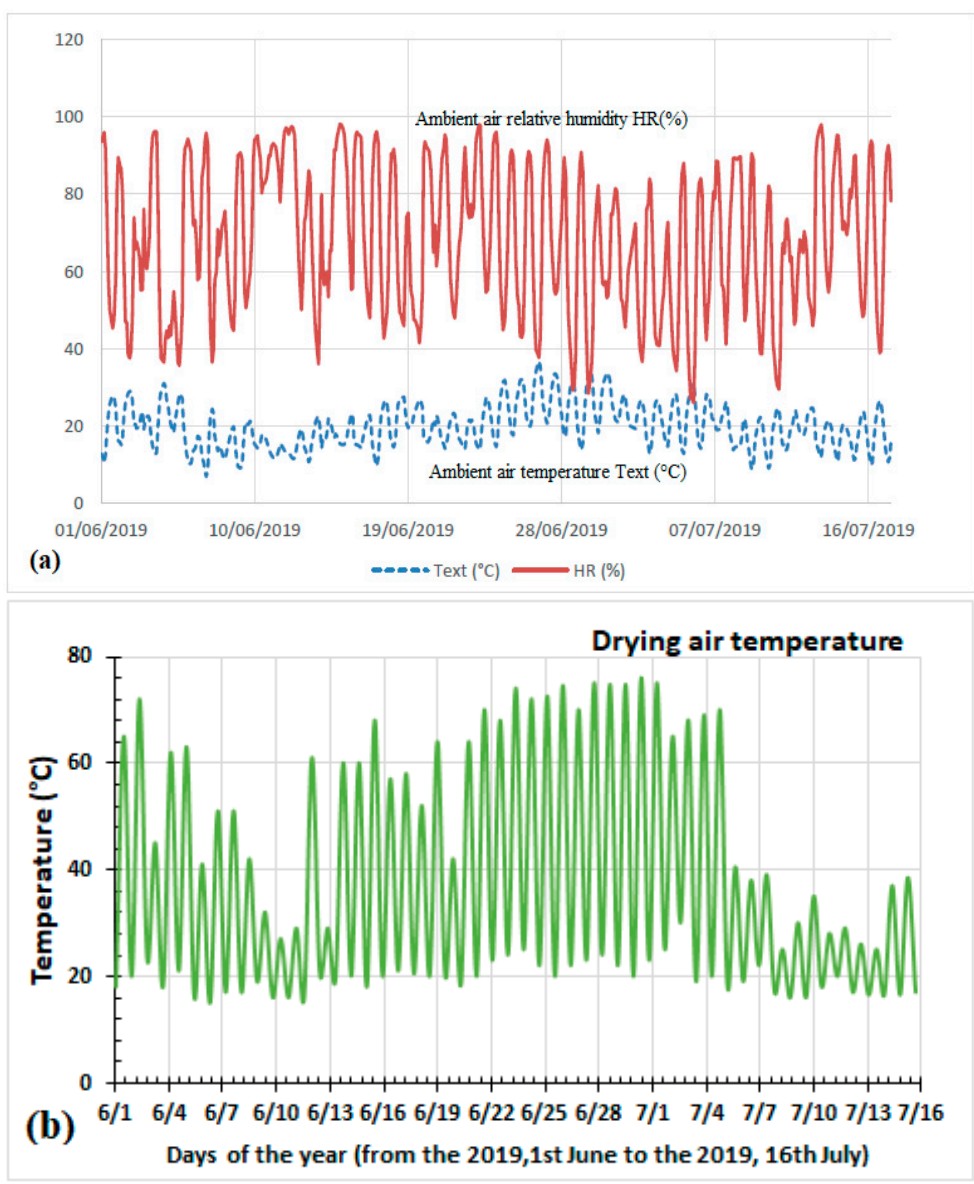

**Figure 4.** Influences of the solar dryer on the characteristics of drying air during six weeks of the spring season in Epinal, from the 1st June 2019 to the 16th July 2019. (**a**) exterior ambient air, (**b**) drying air.

Figure 5 shows the influence of the sun rays on the thermal efficiency of the solar drying during the winter season in Epinal. During the experiment, the door of the drying chamber was closed and drying air was not renewal during the night. Plots in Figure 5 showed that ambient temperature is lower than the drying air temperature both in the night and the day. Also during the day, the difference between drying air temperature and ambient temperature increases when the sun rays increase. The maximum temperature of 40 °C was recorded in the kiln on 5 and 6 January 2020 which was 32 °C higher than the corresponding ambient temperature. Wood temperatures and ambient temperatures were taken from Figure 5. The values of L and Cpb were taken as 2,257,000 J/kg and 1275 J/(K.kg) respectively [26]. The anhydrous mass ($m_o$) used was 28.175 kg, because $m_i$ = 48.05 kg and $H_i$ = 0.7054 kg/kg. After one week, the mass of the logs was equal to 45.6435 kg. The day duration in Epinal during winter is about 9 h. About 28 kWh per cubic meter of wood is needed for fans ventilation according to the literature [12]. Therefore with 0.0251 m$^3$ of wood dried, thus $\sum_j P_j.t_{dj}$ = 2530080 J. The global effective efficiency is equal to $\eta_{gl}$ = 38.71%. This is reasonable when compared with reports of dryers [40].

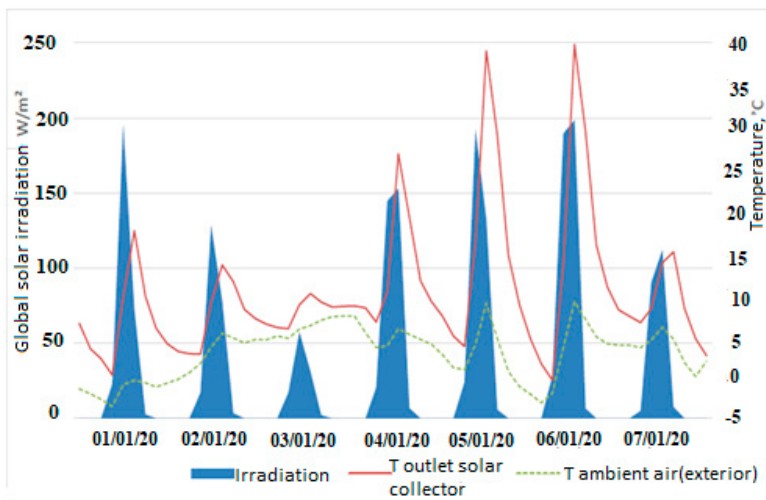

**Figure 5.** Influence of sun rays on the drying air during a week of the winter season in Epinal (Door close and not air renewal).

Maximum thermal efficiency ($\eta_{th}$) is presented in Figure 6 which shows that it is a function of the day and night during winter. In Nigeria, Fuwape and Fuwape [40] found an average thermal efficiency of the wood solar dryer was about 38.5%. Compared to this value, we can say that the present dryer is acceptable.

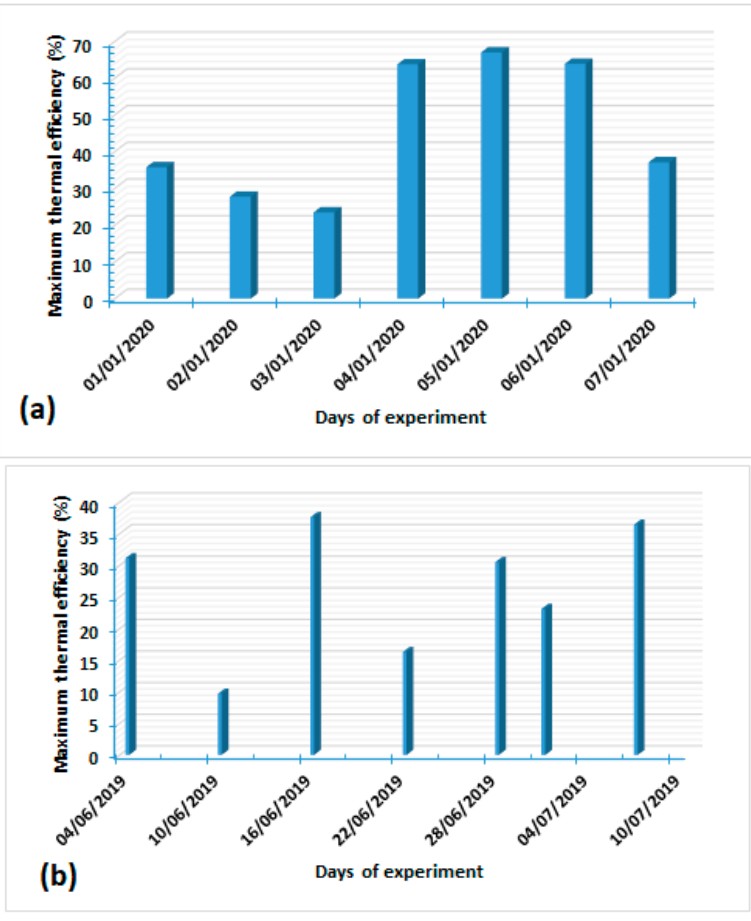

**Figure 6.** The thermal efficiency of the process during the winter season (**a**) and summer season (**b**) in Epinal.

Figure 7 shows the time variation of the relative humidity, ambient temperature and the drying kinetics of the solar drying process in the winter. From Figure 7 it can be seen that the relative humidity of drying air is lower than the one of ambient air (Figure 7a) showing that the drying did not stop. Additionally, Figure 7b shows that the temperatures of the collector and drying chamber of the dryer are higher than the ambient temperature. However from Figure 7c, although the moisture content of logs decreases under the drying conditions, the recommended moisture content required for the logs to be used as fuel was not attained. Thus there is the need for a hybrid dryer with a supplementary electrical heat source, thermal storage material etc. during the winter season. The supplementary heat source will continue the drying process during the off-sunshine period to avoid rewetting of the wood which will prolong the drying periods.

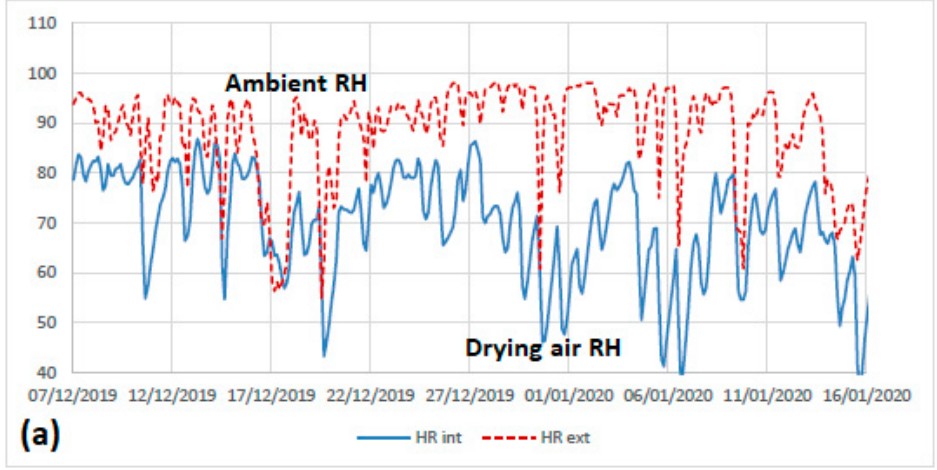

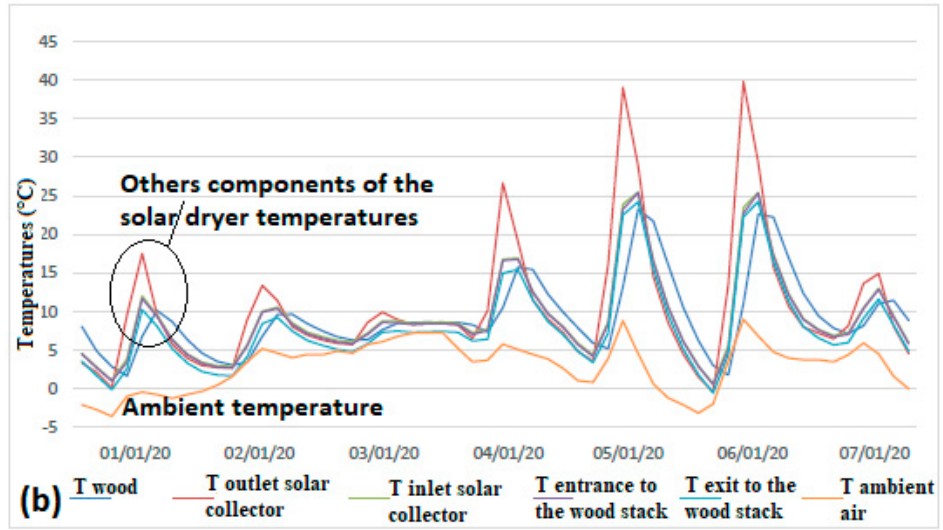

**Figure 7.** *Cont.*

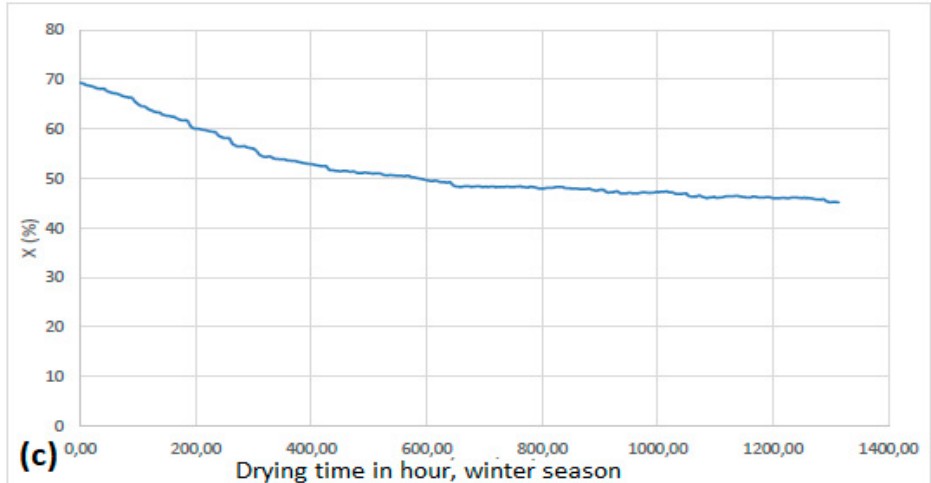

**Figure 7.** Drying kinetics of log fuel during the winter season of experiment in Epinal, (**a**) relative humidity of the ambient air and drying air, (**b**) temperatures of ambient air and internal components, (**c**) moisture content versus drying time.

Figure 8 presents the numerical (model) and experimental moisture content during the winter season 2019/2020 in Epinal. For the validation of the numerical model, experimental drying periods was limited to 50 h as shown in on Figure 7 with a time step used was equal to 1 h (3600 s). The average diameter of the logs has been taken as the thickness (e) and recorded as 0.08425 m with a total log area ($A_b$) of 1.3576 m². The rate of drying air $v_{air}$ was inversely fixed because the positions of logs in the drying chamber doesn't give room to distinguish the real direction of drying air near the logs. Thus, we have retained the value $v_{air}$ as 1.2 m/s. Effectively to achieve good drying in turbulent mode, a value higher than 1 m/s is recommended [41]. Excel 2016 was used to generate all numerical points. From the graph of Figure 8, the evolution of moisture content with time for both the predicted and experimental moisture content followed the same trend as they decreased with time. The coefficient of variations of numerical points with the experimental values given by the model is less than 5% with a mean average error and a mean relative error of 2.33% and 4.53%, respectively. These values are acceptable when compared to those available in the literature [19,25]. The slight differences obtained may be due to the presence of bark on the log, non-uniformity of the diameter of the samples or non-uniformity of the drying air velocity. Thus, mass transfer is given by Equation (1), the global mass transfer coefficient given by Equation (2) and the sorption isotherm model is given by Equation (3a)–(3e) and Equation (4) can be used to model the entire solar dryer using weather conditions of the city.

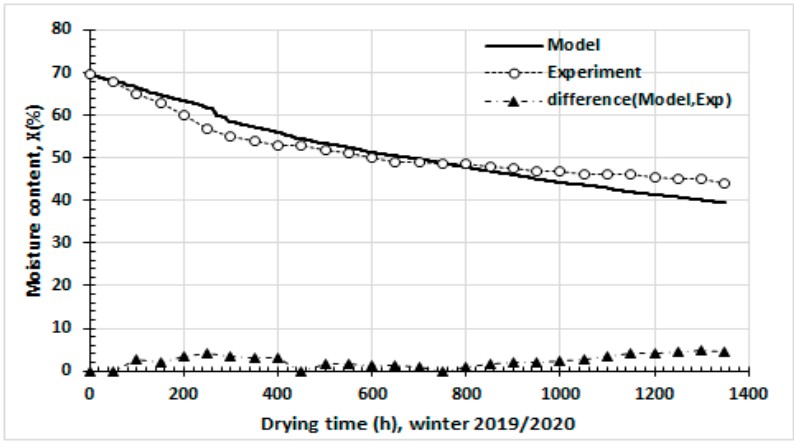

**Figure 8.** Theoretical validation of the drying kinetic.

## 4. Conclusions

The performance of the solar dryer made from the wood panel was presented in this study for drying logs of wood with bark during the winter and spring season in Epinal, France. The solar dryer raised the temperature of the drying air by 38 °C with a global effective efficiency of 39%, which is similar to those obtained in industrial applications. The dryer was able to continue the drying process in the bad weather conditions of the winter season, but it was unable to reach the desired moisture content required for the log to be used as burning fuel under the drying conditions evaluated. Therefore it was recommended that a hybrid mode dryer should be adopted during the winter season to assist the drying process. A model was developed based on the global mass transfer coefficient for wood planks and considering the cylindrical shape of the wood. The average deviation of moisture content between the predicted and experimental values was 2.33%.

**Author Contributions:** Conceptualization, M.S.-T.; methodology, M.S.-T; software, M.S.-T.; validation, M.S.-T., M.C.N. and M.N.A.; formal analysis M.S.-T., M.C.N. and M.N.A.; investigation, M.S.-T., M.C.N. and M.N.A.; resources, M.S.-T.; data curation, M.S.-T.; writing—original draft preparation, M.S.-T.; writing—review and editing, M.S.-T., M.C.N. and M.N.A.; All authors have read and agreed to the published version of the manuscript.

**Funding:** The principal author acknowledges the International Tropical Timber Organization (ITTO) for financially supporting a part of this work (ITTO Ref. Number: 022/19A).

**Acknowledgments:** The first author gratefully acknowledges this third financial support and is thankful to Angélique Leonard of the University of Liege (Belgium), André Zoulalian and Yann Rogaume of the University of Lorraine (France) for positive letters of reference that facilitated my grant. Merlin Simo-Tagne is also grateful to Romain Rémond, Tristan Stern, Lucas Delaunay and Tom Bourgault of LERMaB (Epinal) for precious help in completing the experimental results. This paper is dedicated to the memory of Beguide Bonoma of the University of Yaoundé I (Cameroon) who supervised my PhD thesis and died 29 March 2020.

**Conflicts of Interest:** The authors declare no conflict of interest.

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
