# Peer review of "Experimental Modelling of a Solar Dryer for Wood Fuel in Epinal (France)"

_2673-3951, doi:10.3390/modelling1010003_

Round 1

Reviewer 1 Report

The definitions of the temperatures in Figure 7b are not clear.

Under section 2.1 Material, there is mention of "east side", "west side" and "north side" of the dryer with reference to Fig.1. From the figure, it is difficult to know which side is east, west or north without arrows on the figure showing these directions. Another way of referring to these sides could be used; for instance side A, with a label on that side of the figure.

Suggestions for Grammar are in the file "Reviewer comments" attached.

Author Response

ANSWER : All the grammer corrections listed below have been corrected and highlighted in red ink

72 heat energy while 32 kWh/m3 (28 kWh/m3) for solar kiln driers (of electrical energy for ventilation)

Comment: Recast, make it clear which figure is for solar kiln driers and which one is for electrical ventilation.

82 used in Moroccan climate. Although they had a satisfactory performance but it seemed unreliable

97 The wood stack dimensions were 2.16 x 1.60 x 2.5 m with a distance of 0.027 m between layers. The They

99 quality on the drying time. Seasons, renewal rate and mass storage where were more influenced by the

105 The literatures reviewed above does not only gave give insight on the parameters for the construction

117 under a transparent cover served as the solar collector absorber that helps to absorb a greater part of solar

118 irradiation radiation during a summer day. In between the cover and the collector is a layer of air that permits

122 a good movement of air under the black panel at an average airflow of 28 m3/h. In-betweens between this

2190temperature of the drying chamber when the door is close closed during the six weeks of the spring season

239 32°C higher than the corresponding ambient temperature. One A one week drying periods were used for

262 content of logs decreases, but it is impossible to reach the recommended moisture content so as it can

292 compared to those present in the industry. This dryer was able to satisfactorily continued continue the drying

Reviewer 2 Report

In section 1.

  • Line 65 '' Application of solar in drying some products gives acceptable drying kinetics''. Could you name these application and products and add some numbers or percentages?

  • Line 81, Line 87, Could you add authors names in the beginning of the sentence.

  • In the last paragraph, Line 105, please stat the aim of the study. Also clarify the novelty of the model you have used compared to those that other researchers have used in the past.

In section 2.

  • Line 151, '' the payback is after 3 years and 3 months''. Please add the reference that you compared the model with economically.

In section 3

  • Line 224, the figure is not clear and without title or description.

  • Line 227, figure 4 required to be clear.

  • Line 233, the description of figure 5 required to be prior the figure. Please use the same note with all figures in the article.

In section 5

  • The outcome of the article need to be linked to the aim in the introduction.

Author Response

In section 1.

Line 65 '' Application of solar in drying some products gives acceptable drying kinetics''. Could you name these application and products and add some numbers or percentages?

Answer : we showed a reference at the end of the work where this applications can be found

 Line 81, Line 87, Could you add authors names in the beginning of the sentence.

Answer : This section is re-written and highlighted in red

 In the last paragraph, Line 105, please stat the aim of the study. Also clarify the novelty of the model you have used compared to those that other researchers have used in the past.

Answer : This section is re-written and highlighted in red

In section 2.

 Line 151, '' the payback is after 3 years and 3 months''. Please add the reference that you compared the model with economically.

Answer : It is okay now, Ref. 28.

In section 3

Line 224, the figure is not clear and without title or description.

Answer : It is okay now.

Line 227, figure 4 required to be clear.

Answer : It is okay now.

Line 233, the description of figure 5 required to be prior the figure. Please use the same note with all figures in the article.

Answer : We improve now the quality of this figure.

In section 5

The outcome of the article need to be linked to the aim in the introduction.

Answer : This has been  re-written and highlighted in red

Reviewer 3 Report

The manuscript should be improved and, at least, the following changes should be carried out:

1) More details about the used sensors and their uncertainties must be added in the text. For instance, how did you measure the received solar irradiance by the dryer?

2) Caption in figure 7 (b) should be corrected and re-written in English.

3) Line 263: More comment are needed about the possibility of using hybrid dryers during cold periods. What about heat storage process?

4) For the validation section, the MAE and the RMSE should be presented. In addition, the authors must give scientific reasons (major and manor reasons) about the discrepancy between measured and predicted results.

5) The sentence “modeling of a solar dryer” should be avoided in the text. The authors have modeled only the wood drying kinetic and not the solar dryer system!.

Author Response

The manuscript should be improved and, at least, the following changes should be carried out:

1) More details about the used sensors and their uncertainties must be added in the text. For instance, how did you measure the received solar irradiance by the dryer?

Answer : Solar irradiation valuess was obtained in the weather station of the laboratory, near the studied dryer. Pt100 sensors have been used with a square of correlation equal to 0.9999. This precision has been added in the manuscript.

2) Caption in figure 7 (b) should be corrected and re-written in English.

Answer : It has been corrected

3) Line 263: More comment are needed about the possibility of using hybrid dryers during cold periods. What about heat storage process?

Answer : This  has been expanded and shown in red ink

4) For the validation section, the MAE and the RMSE should be presented. In addition, the authors must give scientific reasons (major and manor reasons) about the discrepancy between measured and predicted results.

Answer : It has been corrected

5) The sentence “modeling of a solar dryer” should be avoided in the text. The authors have modeled only the wood drying kinetic and not the solar dryer system!.

Answer : Dear Reviewer, this work presents an experimental modeling of our solar solar dryer based on global mass transfer coefficient.  The work showed that  the global mass transfer coefficient used by other authors can be used to model the drying kinetic of wood.

Reviewer 4 Report

  1. The abstract must be rewritten to emphasize the novelty of the work.
  2. Equations must be re organized and typed in a better format.
  3. Nomenclature is missing which can make it easier to understand the equations.
  4. Figures 3 and 5, must be in better resolution and have a consistent font with the body of the manuscript.
  5. Figure 4, the numbers are not clear and even some of them are difficult to read.
  6. 16 references out of 40 references are from the authors, which is too much for self-citing.

Author Response

  1. The abstract must be rewritten to emphasize the novelty of the work.

Answer : It has been corrected

  1. Equations must be re organized and typed in a better format.

Answer : It has been corrected

  1. Nomenclature is missing which can make it easier to understand the equations.

Answer : It has been corrected

  1. Figures 3 and 5, must be in better resolution and have a consistent font with the body of the manuscript.

Answer : It has been corrected

  1. Figure 4, the numbers are not clear and even some of them are difficult to read.

Answer : It has been corrected

  1. 16 references out of 40 references are from the authors, which is too much for self-citing.

Answer : Dear Reviewer, this work is the  consequence of all our previous works from many years. The authors has worked extensively in this area and has continued to expand their research in this area which reflected the references.

Round 2

Reviewer 4 Report

It is acceptable in the current format